# Simulation of Diffusion Bonding of Different Heat Resistant Nickel-Base Alloys

**Albert R. Khalikov** [1] , **Evgeny A. Sharapov** [2], **Vener A. Valitov** [3,4] , **Elvina V. Galieva** [3] ,
**Elena A. Korznikova** [5,6] and **Sergey V. Dmitriev** [5,6,*]

[1] The Faculty of Avionics, Energy Engineering and Infocomm Technology, Ufa State Aviation
Technical University, K. Marx St., 12, 450000 Ufa, Russia; khalikov.albert.r@gmail.com
[2] LLC Bashneft Polus, Chernyshevsky St., 60, 450076 Ufa, Russia; ufa_usinsk@mail.ru
[3] Institute for Metals Superplasticity Problems, Russian Academy of Sciences, Khalturin St., 39,
450001 Ufa, Russia; valitov_va@mail.ru (V.A.V.); galieva_elvina_v@mail.ru (E.V.G.)
[4] Bashkir State University, 32 Zaki Validi St., 450076 Ufa, Russia
[5] Institute for Molecule and Crystal Physics, Ufa Federal Research Centre of the Russian Academy of Sciences,
Oktyabrya Av., 151, 450075 Ufa, Russia; elena.a.korznikova@gmail.com
[6] Faculty of Mechanics, Ufa State Petroleum Technological University, Kosmonavtov St. 1, 450062 Ufa, Russia
* Correspondence: dmitriev.sergey.v@gmail.com

**Abstract:** Currently, an important fundamental problem of practical importance is the production
of high-quality solid-phase compounds of various metals. This paper presents a theoretical model
that allows one to study the diffusion process in nickel-base refractory alloys. As an example,
a two-dimensional model of ternary alloy is considered to model diffusion bonding of the alloys with
different compositions. The main idea is to divide the alloy components into three groups: (i) the
base element Ni, (ii) the intermetallic forming elements Al and Ti and (iii) the alloying elements.
This approach allows one to consider multi-component alloys as ternary alloys, which greatly
simplifies the analysis. The calculations are carried out within the framework of the hard sphere
model when describing interatomic interactions by pair potentials. The energy of any configuration
of a given system is written in terms of order parameters and ordering energies. A vacancy diffusion
model is described, which takes into account the gain/loss of potential energy due to a vacancy jump
and temperature. Diffusion bonding of two dissimilar refractory alloys is modeled. The concentration
profiles of the components and order parameters are analyzed at different times. The results obtained
indicate that the ternary alloy model is efficient in modeling the diffusion bonding of dissimilar
Ni-base refractory alloys.

**Keywords:** diffusion welding; diffusion; metal; refractory alloy; Monte Carlo method

---

## 1. Introduction

The theory and practice of diffusion welding of dissimilar metals has been actively developed
over the past decades due to the undoubted importance of this process for various technologies [1,2].
Some alloys can be successfully bonded without the use of an interlayer [3–5]. However, in most cases,
to avoid the formation of brittle intermetallic inclusions, an interlayer metal is used. Examples are
bonding of titanium to steel [6–9], nickel [10], copper [11] and the use of copper interlayer for bonding
stainless steel to Zircaloy 4 [12]. Heat-resistant intermetallic alloys can have unique physical and
mechanical properties (heat resistance, resistance to oxidation, corrosion, creep performance, etc.) that
makes them ideal for a number of applications in aerospace industry [13–17]. Structure and properties
of Ni-base alloys can be improved by laser treatment [18,19] and thermomechanical treatment [20,21].
Electro-discharge machining process of Inconel-718 superalloy has been optimized in [22].

Interest in these alloys is due to the possibility of achieving higher alloy characteristics. Intermetallic alloys based on $Ni_3Al$ are used in parts of combustion chambers of aircraft engines with operating temperatures up to 1300 °C. Methods for joining metals and superalloys include diffusion welding [15,23], fusion welding, friction welding [24–28], diffusion welding in a liquid phase [29], friction stir welding [28], etc. When dissimilar metals are joined together in solid phases, undesirable brittle intermetallic phases are often formed, which reduce the mechanical properties of parts [3,6,7,12,30].

One of the ways to predict the properties of intermetallic phases is computer modeling of the structure of alloys, which allows to reveal undesirable structural states for a given alloy prior to experiment. Various computational methods can be used, for example, molecular dynamics (MD) [31], phase-field [32], finite element [33], Monte Carlo [34–37] and others methods. Diffusion processes have also been modeled in frame of two-dimensional continuum model [38]. It is difficult to apply MD to simulate the diffusion process, since the time scale achieved by this method is on the order of 1 ns, and diffusion bonding takes about 15 min. The time difference is huge, twelve orders of magnitude. This is why the development of the Monte Carlo method is very important. The use of continuum models for diffusion in multi-component media is not straightforward, since it requires knowledge of the matrix of diffusion coefficients [39]. Various simulation methods work at different scales and they make it possible to establish ways to achieve the required alloy structure, to trace the formation of intermetallic inclusions and their distribution over the volume and to study the effect of alloying and temperature changes. Modeling the ordering process of alloys makes it possible to establish an equilibrium structure and calculate its parameters, such as energy, degree of short-range and long-range order, depending on the concentration of components.

The main difficulty in atomistic simulation of heat-resistant Ni-base alloys is that they include many alloying elements. This prevents the use of MD modeling due to the complexity of constructing interatomic potentials. It is necessary to adopt additional simplifying assumptions to create models describing diffusion processes.

In this work, within the framework of a two-dimensional model, the structure of heat-resistant VKNA-25 and EP975 alloys is investigated. These alloys are based on the $Ni_3Al$ intermetallic compound containing alloying elements with different concentrations. Alloy components are divided into three groups depending on the role they play in the alloy. The Monte Carlo algorithm is used to simulate the diffusion process by the vacancy mechanism in a three-component metallic systems.

In Section 2, we describe the experimental results on diffusion bonding of VKNA-25 and EP975 alloys, present the derivation of order parameters for ternary alloys following [37], describe the three-component alloy model and the vacancy diffusion model. Then, in Section 3, the numerical results and their discussion are presented. Section 4 concludes this work.

## 2. Experimental Results and Simulation Setup

### 2.1. Experimental Results

The chemical composition of VKNA-25 and EP975 alloys is shown in Table 1 in weight percent [40]. Both alloys are based on nickel, while aluminum and titanium are elements that form the intermetallic phase $Ni_3Al$, where aluminum atoms can be replaced by titanium atoms. Many alloying elements are also present.

Figure 1 shows the microstructures of (a) VKNA-25 and (b) EP975 alloys. In (c), the microstructure of the solid phase joint of the two alloys is presented. In the images, the $\gamma'$ phase, which is the $Ni_3Al$ intermetallic compound, has a dark color, and the solid solution of alloying elements in $Ni_3Al$ is the $\gamma$ phase, which has a light color. The microstructures of the alloys and the joint zone were studied by scanning electron microscope Mira 3LMH (TESCAN, Brno , Czech Republic).

**Table 1.** Chemical composition of VKNA-25 and EP975 alloys in weight percent [40].

| Atomic Element | VKNA-25 | EP975 |
|---|---|---|
| Al | 8.1–8.6 | 4.5–5.1 |
| Ti | 0.3–0.7 | 2.0–2.7 |
| Cr | 5.6–6.0 | 7.5–9.0 |
| W | 2.5–3.5 | 9.5–11.0 |
| Mo | 4.5–5.5 | 0.8–1.5 |
| Nb | – | 1.0–2.0 |
| Co | 4.0–5.0 | 14.1–17.0 |
| Re | 1.2–1.6 | – |
| C | 0.005–0.02 | 0.1–0.16 |
| Ni | Bal. | Bal. |

EP975 alloy is a wrought polycrystalline alloy containing 55% of the hardening $\gamma'$ phase, and VKNA-25 alloy is a cast intermetallic alloy with a single-crystal structure based on Ni$_3$Al, in which the content of the intermetallic $\gamma'$ phase is 85–90% [15,16].

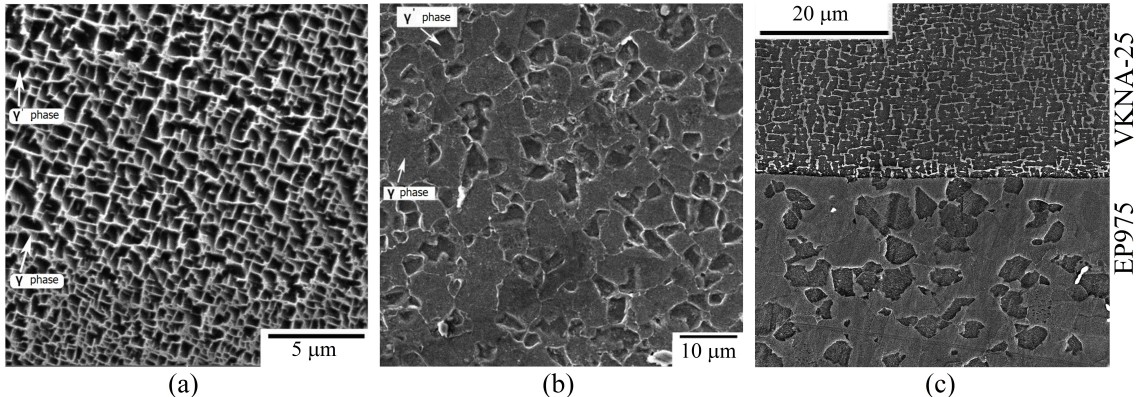

**Figure 1.** Microstructure of the intermetallic alloys VKNA-25 (**a**) and EP975 (**b**). Both alloys are the mixture of the Ni$_3$Al intermetallic compound ($\gamma'$ phase), which appears in dark color, and the solid solution of alloying elements in Ni$_3$Al ($\gamma$ phase) has a light color. (**c**) Microstructure of the joint of EP975 (bottom) and VKNA-25 (top) alloys.

Cylindrical samples of VKNA-25 and EP975 alloys were 20 mm in diameter and 30 mm in height. Pressure welding of VKNA-25 and EP975 alloys was performed in vacuum and welding temperature was 1125 °C. The two cylindrical samples were put in contact and compressed with the strain rate of $\dot{\varepsilon} = 10^{-4}$ s$^{-1}$. Note that VKNA-25 is a hard-to-deform single-crystal alloy, while EP975 is polycrystalline alloy that flows superplastically at the selected temperature and strain rate. The superplasticity of EP975 alloy is important because it helps to achieve a better physical contact during pressure welding and to minimize the formation of voids. Strain control loading is performed so that pressure is not a controlled parameter. A compressive strain is applied normal to the interface to achieve better physical contact between the parts to be welded.

The microstructure of the joint is presented in Figure 1c, where VKNA-25 and EP975 alloys are above and below the contact plane, respectively. A high-quality joint is obtained with no voids or other imperfections. Close to the interface, the $\gamma$ phase in VKNA-25 alloy has lighter color than in the bulk. This indicates a change in the composition of the alloys in the vicinity of the contact plane. This is confirmed by the components concentration profiles in the vicinity of the contact plane presented in Figure 2a. The distribution of elements in the solid-phase joint zone was determined by the X-ray energy-dispersive spectroscopy (EDS) on a Tescan VEGA 3SBH scanning electron microscope with an EDX attachment. The measurements were carried out only from the $\gamma$-phase, moving from the EP975 alloy to the VKNA-25 alloy with a step of 10 μm. The diffusion zone was found to be about 35 μm

wide. To plot the distribution of alloying elements, the EDS data were averaged over at least three measurements per point. In Figure 2b, the concentrations of three groups of alloy components are shown: the base element Ni, the intermetallic-forming Al and Ti are combined and named Al and all other elements are combined and named Cr. The merit of combining alloy elements into these three groups is explained in Section 2.4.

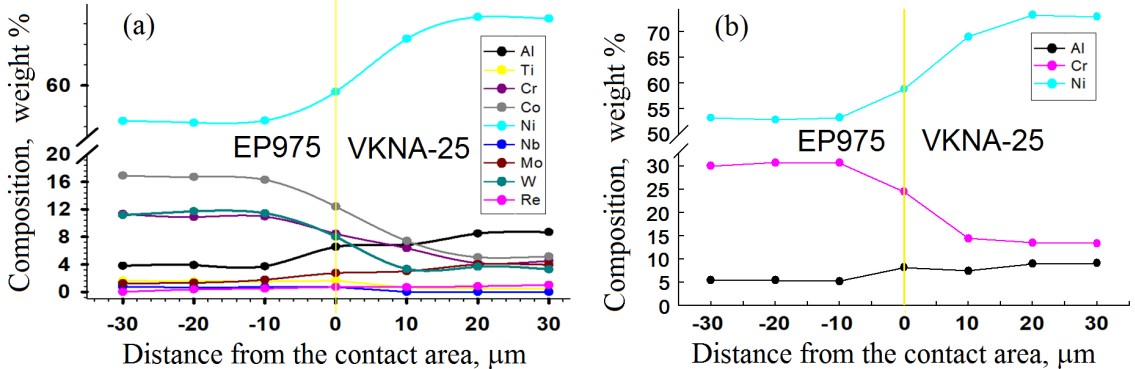

**Figure 2.** Variations in the composition of alloys near the joint. (**a**) Results of measurements. (**b**) Composition for three groups of elements: the base element Ni, the intermetallic-forming Al and Ti are combined and named Al and all other elements are combined and named Cr.

## 2.2. Order Parameters for Ternary Alloy

Multi-component alloys can contain domains of ordered and disordered phases, as well as segregation of pure components. It is important to quantify the degree of ordering and the content of segregations. In this section, the order parameters for ternary alloys derived in [37] are reproduced for the sake of the readers. In addition, a new result is presented—the derivation of the parameters that determine the proximity of the alloy to decomposition into pure components.

All the expressions presented below are valid for any crystal lattice of any dimension. The information about the type and dimensionality of the lattice is introduced through the coordination numbers $N_i$, i.e., the numbers of atoms on the $i$-th coordination sphere.

Let us consider a three-component alloy of composition $A_m B_n C_k$, in which atoms of types $A$, $B$ and $C$ are located at lattice sites. The lattice has $N_i$ atoms on the $i$-th coordination sphere. The concentrations of atoms of types $A$, $B$ and $C$ are equal to, respectively,

$$c_A = \frac{m}{m+n+k}, \quad c_B = \frac{n}{m+n+k}, \quad c_C = \frac{k}{m+n+k}. \tag{1}$$

We denote by $p_{KL}^{(i)}$ the probability that the $i$-th coordination sphere of an atom of the type $K$ contains an atom of the type $L$, where $K, L = \{A, B, C\}$. In a ternary alloy, the following relationships exist between the nine probabilities $p_{KL}^{(i)}$, and the three concentrations $c_A, c_B, c_C$:

$$p_{AA}^{(i)} + p_{AB}^{(i)} + p_{AC}^{(i)} = 1, \quad p_{BA}^{(i)} + p_{BB}^{(i)} + p_{BC}^{(i)} = 1, \quad p_{CA}^{(i)} + p_{CB}^{(i)} + p_{CC}^{(i)} = 1, \quad c_A + c_B + c_C = 1, \tag{2}$$

$$c_A \left( p_{AA}^{(i)} - c_A \right) = -\frac{1}{2} \left[ c_A \left( p_{AB}^{(i)} + p_{AC}^{(i)} \right) + c_B \left( p_{BA}^{(i)} - 2c_A \right) + c_C \left( p_{CA}^{(i)} - 2c_A \right) \right], \tag{3}$$

$$c_B \left( p_{BB}^{(i)} - c_B \right) = -\frac{1}{2} \left[ c_B \left( p_{BA}^{(i)} + p_{BC}^{(i)} \right) + c_A \left( p_{AB}^{(i)} - 2c_B \right) + c_C \left( p_{CB}^{(i)} - 2c_B \right) \right], \tag{4}$$

$$c_C \left( p_{CC}^{(i)} - c_C \right) = -\frac{1}{2} \left[ c_C \left( p_{CA}^{(i)} + p_{CB}^{(i)} \right) + c_A \left( p_{AC}^{(i)} - 2c_C \right) + c_B \left( p_{BC}^{(i)} - 2c_C \right) \right]. \tag{5}$$

Let us denote by $\varphi_{KL}^{(i)}$ the binding energy of a pair of atoms of types $K$ and $L$ located at a distance equal to the radius of the $i$-th coordination sphere.

The potential energy of the structure per atom, taking into account the interaction of atoms in the first $I$ coordination spheres, can be written in the form

$$E = \frac{1}{2}\sum_{i=1}^{I} N_i \left( \begin{array}{c} c_A(p_{AA}^{(i)}\varphi_{AA}^{(i)} + p_{AB}^{(i)}\varphi_{AB}^{(i)} + p_{AC}^{(i)}\varphi_{AC}^{(i)}) \\ +c_B(p_{BA}^{(i)}\varphi_{BA}^{(i)} + p_{BB}^{(i)}\varphi_{BB}^{(i)} + p_{BC}^{(i)}\varphi_{BC}^{(i)}) \\ +c_C(p_{CA}^{(i)}\varphi_{CA}^{(i)} + p_{CB}^{(i)}\varphi_{CB}^{(i)} + p_{CC}^{(i)}\varphi_{CC}^{(i)}) \end{array} \right). \tag{6}$$

The energy of the completely disordered state of the structure is determined by Equation (6) for $p_{AA}^{(i)} = p_{BA}^{(i)} = p_{CA}^{(i)} = c_A$, $p_{AB}^{(i)} = p_{BB}^{(i)} = p_{CB}^{(i)} = c_B$, $p_{AC}^{(i)} = p_{BC}^{(i)} = p_{CC}^{(i)} = c_C$, which gives

$$E_{\text{disord}} = \frac{1}{2}\sum_{i=1}^{I} N_i \left( c_A^2\varphi_{AA}^{(i)} + c_B^2\varphi_{BB}^{(i)} + c_C^2\varphi_{CC}^{(i)} + 2c_Ac_B\varphi_{AB}^{(i)} + 2c_Ac_C\varphi_{AC}^{(i)} + 2c_Bc_C\varphi_{BC}^{(i)} \right). \tag{7}$$

The energy of decomposition into pure components is determined by Equation (6) for $p_{AB}^{(i)} = p_{BA}^{(i)} = p_{AC}^{(i)} = p_{CA}^{(i)} = p_{BC}^{(i)} = p_{CB}^{(i)} = 0$, $p_{AA}^{(i)} = 1$, $p_{BB}^{(i)} = 1$, $p_{CC}^{(i)} = 1$, which gives

$$E_{\text{decomp}} = \frac{1}{2}\sum_{i=1}^{I} N_i \left( c_A\varphi_{AA}^{(i)} + c_B\varphi_{BB}^{(i)} + c_C\varphi_{CC}^{(i)} \right). \tag{8}$$

Let us choose the $E_{\text{disord}}$ energy as a reference point, and characterize the energy of any structure by the difference

$$\triangle E_1 = E - E_{\text{disord}} = \frac{1}{4}\sum_{i=1}^{I} N_i \left( \alpha_{AB}^{(i)}\omega_{AB}^{(i)} + \alpha_{AC}^{(i)}\omega_{AC}^{(i)} + \alpha_{BC}^{(i)}\omega_{BC}^{(i)} \right), \tag{9}$$

where we have introduced the order parameters

$$\alpha_{AB}^{(i)} = c_A p_{AB}^{(i)} + c_B p_{BA}^{(i)} - 2c_Ac_B, \quad \alpha_{AC}^{(i)} = c_A p_{AC}^{(i)} + c_C p_{CA}^{(i)} - 2c_Ac_C, \quad \alpha_{BC}^{(i)} = c_B p_{BC}^{(i)} + c_C p_{CB}^{(i)} - 2c_Bc_C, \tag{10}$$

and ordering energies

$$\omega_{AB}^{(i)} = \varphi_{AA}^{(i)} + \varphi_{BB}^{(i)} - 2\varphi_{AB}^{(i)}, \quad \omega_{AC}^{(i)} = \varphi_{AA}^{(i)} + \varphi_{CC}^{(i)} - 2\varphi_{AC}^{(i)}, \quad \omega_{BC}^{(i)} = \varphi_{BB}^{(i)} + \varphi_{CC}^{(i)} - 2\varphi_{BC}^{(i)}. \tag{11}$$

When deriving Equation (9), Equations (2)–(5) were used.

On the other hand, if the $E_{\text{decomp}}$ energy is used as a reference point, then the energy difference is

$$\triangle E_2 = E - E_{\text{decomp}} = \sum_{i=1}^{I} \frac{N_i}{4} \left[ \beta_{AB}^{(i)}\omega_{AB}^{(i)} + \beta_{AC}^{(i)}\omega_{AC}^{(i)} + \beta_{BC}^{(i)}\omega_{BC}^{(i)} \right], \tag{12}$$

where a set of parameters defining the proximity of the alloy to decomposition into pure components is introduced

$$\beta_{AB}^{(i)} = c_A p_{AB}^{(i)} + c_B p_{BA}^{(i)}, \quad \beta_{AC}^{(i)} = c_A p_{AC}^{(i)} + c_C p_{CA}^{(i)}, \quad \beta_{BC}^{(i)} = c_B p_{BC}^{(i)} + c_C p_{CB}^{(i)}. \tag{13}$$

When deriving Equation (12), we used Equations (2)–(5).

As we can see from Equation (9), the deviation of the energy of any structure from the energy of a disordered alloy is uniquely determined by the coordination numbers $N_i$, ordering energies $\omega_{KL}^{(i)}$ and order parameters $\alpha_{KL}^{(i)}$. Similarly, Equation (12) defines the deviation of the energy of any structure from the energy of the alloy decomposed into pure components in terms of the parameters $\beta_{KL}^{(i)}$ and ordering energies $\omega_{KL}^{(i)}$.

For a two-component alloy ($c_C = 0$), the above equations are greatly simplified. For example, Equations (2)–(5) reduce to

$$p_{AA}^{(i)} + p_{AB}^{(i)} = 1, \quad p_{BA}^{(i)} + p_{BB}^{(i)} = 1, \quad c_A(p_{AA}^{(i)} - c_A) = c_B(p_{BB}^{(i)} - c_B), \tag{14}$$

and Equation (9) assumes the form

$$\triangle E = E - E_{\text{disord}} = \frac{1}{4} \sum_{i=1}^{I} N_i \alpha_{AB}^{(i)} \omega_{AB}^{(i)}. \tag{15}$$

On the other hand, the complexity of the relationships between the probabilities $p_{KL}^{(i)}$ increases very rapidly with an increase in the number of alloy components. For this reason, in the following section we propose a ternary alloy model that can be used to describe diffusion processes in multi-component refractory nickel-base alloys.

### 2.3. Vacancy Diffusion Model

Let us describe the mathematical model of the diffusion process in the alloy according to the vacancy mechanism in the model of hard spheres, which can be applied to a multi-component alloy of the composition, specified on a lattice of any type and any dimension. In this model, it is assumed that an elementary act of diffusion is the transition of one of the atoms surrounding a vacancy to its place. It is assumed that any atom from the first $K$ coordination spheres can jump into the vacant lattice site. The number of such atoms is $M = \sum_{i=1}^{K} N_i$, where $N_i$ is the coordination number for the $i$-th coordination sphere. Each of the $M$ atoms is assigned the probability $p_m$, $m = 1, ..., M$, to occupy the place of the vacancy in an elementary act of diffusion, so that $\sum_{m=1}^{M} p_m = 1$. For this purpose, the change in the energy of the alloy, $\Delta E_m$, associated with the transition of the $m$-th atom to a vacant place for a given temperature of the alloy $T$ is calculated. The required probabilities are determined as follows

$$p_m = \frac{P_m}{\sum_{m=1}^{M} P_m}, \tag{16}$$

where

$$P_m = \exp\left(-\frac{\Delta E_m}{kT}\right), \tag{17}$$

and $k = 8.61733 \times 10^{-5}$ eV/K is the Boltzmann constant.

Note that for increasing temperature the probabilities $p_m$ will tend to become equal, regardless the values of the energies $\Delta E_m$, and this will lead to the disordered state. On the other hand, for relatively small temperatures, the probabilities $p_m$ will depend on $\Delta E_m$, leading to the ordering process. Pressure is not taken into account in this model, because we use the rigid lattice assumption without taking into account the effect of atomic relaxation.

### 2.4. Ternary Alloy Model and Simulation Setup

For modeling multi-component refractory alloys, we divide all elements into three groups. The first group includes a single element, nickel, which is the base element. The second group combines the intermetallic-forming elements aluminum and titanium. Finally, the third group includes all alloying elements. We will call these groups according to their main elements, that is, the first group is $A = $ Ni, the second is $B = $ Al and the third one is $C = $ Cr. It should be noted that the ternary alloy model can describe the considered multi-component superalloys only at a qualitative level.

In Table 2, we present the content of the three components in EP975 and VKNA-25 alloys in weight percent and atomic percent.

**Table 2.** Three groups of elements: constituents of groups, mass and atomic percentages of elements of VKNA-25 and EP975 alloys.

| Group Notation | Group Description | EP975 | | VKNA-25 | |
|:---:|:---:|:---:|:---:|:---:|:---:|
| | | wt.% | at.% | wt.% | at.% |
| Ni | Base (Ni) | 60.04 | 54.8 | 71.185 | 63.7 |
| Al | Intermetallic-forming elements (Al and Ti) | 7.1 | 14.0 | 8.9 | 16.5 |
| Cr | Alloying elements (all other elements) | 32.86 | 31.8 | 19.915 | 19.8 |

In our experiments and in our model, we do not have solidification because joining takes place in the solid state without melting. The high temperature activates the diffusion and superplastic flow of the EP975 alloy, and these two factors play a positive role in achieving a high-quality joint.

The main assumptions made in the model are the approximation of pair interatomic interactions and the approximation of a rigid crystal lattice, i.e., the assumption that the atoms occupy the sites of undeformed crystal lattice, or, in other words, the effects of atomic relaxation are not taken into account.

Our simulation will be carried out with the use of the toy model. We consider two-dimensional square lattice with the interatomic distance equal to unity, which can be always achieved by a proper choice of the unit of length. The atoms in the computational cell are numbered by the indices $1 \leq n_x \leq N_x$ and $1 \leq n_y \leq N_y$ in the horizontal and vertical directions, respectively. The size of the computational cell is $N_x \times N_y = 200 \times 400$ and periodic boundary conditions are applied. The typical interatomic distance in metals is about 2.5 Å, then the size of the computational cell is about 0.1 μm. This is two orders of magnitude smaller than the experimentally observed diffusion zone of 35 μm, see Figure 2. An increase in the size of the computational cell requires an increase in the simulation time.

To describe interatomic interactions, for simplicity, the Morse pair potential is chosen

$$\varphi_{KL}(r^{(i)}) = D_{KL} \left\{ \exp\left[-2\theta_{KL}(r^{(i)} - R_{KL})\right] - 2\exp\left[-\theta_{KL}(r^{(i)} - R_{KL})\right] \right\}, \quad (18)$$

where $D_{KL}$, $\theta_{KL}$ and $R_{KL}$ are the parameters of the potential for the atoms of sorts $K$ and $L$, and $r^{(i)}$ is the radius of the $i$-th coordination sphere. The following values of the potential parameters are chosen:

$$
\begin{aligned}
R_{\text{NiNi}} &= 1.05, & \theta_{\text{NiNi}} &= 5.0, & D_{\text{NiNi}} &= 3.2; \\
R_{\text{AlAl}} &= 1.05, & \theta_{\text{AlAl}} &= 5.0, & D_{\text{AlAl}} &= 3.8; \\
R_{\text{CrCr}} &= 1.05, & \theta_{\text{CrCr}} &= 5.0, & D_{\text{CrCr}} &= 3.0; \\
R_{\text{NiAl}} &= 1.05, & \theta_{\text{NiAl}} &= 5.0, & D_{\text{NiAl}} &= 4.4; \\
R_{\text{NiCr}} &= 1.05, & \theta_{\text{NiCr}} &= 5.0, & D_{\text{NiCr}} &= 3.15; \\
R_{\text{AlCr}} &= 1.05, & \theta_{\text{AlCr}} &= 5.0, & D_{\text{AlCr}} &= 3.45.
\end{aligned} \quad (19)
$$

As it can be seen from Equation (19), for simplicity, we take equal values of the equilibrium distance $R_{KL}$, which reflects the fact that we do not take into account atomic relaxation effects. The equilibrium distance is slightly (5%) greater than the interatomic distance, which is always so when long-range interactions are considered. In our simulations, three coordination shells are taken into account. We also take equal values for the parameters $\theta_{KL}$, that define the rigidity of bonds, and the reason is the same, the lattice is assumed to be non-deformable. The depth of the potentials (binding energies) are chosen in a way to obtain the desired structure of the alloys. In particular, we take $D_{\text{NiAl}}$ larger than $D_{\text{NiNi}}$ and $D_{\text{AlAl}}$ to get negative ordering energy $\omega_{\text{NiAl}} = \varphi_{\text{NiNi}} + \varphi_{\text{AlAl}} - 2\varphi_{AB}$, so that the formation of the intermetallic compound $Ni_3Al$ is energetically favorable. The Cr-Cr potential is the shallowest one in order to prevent segregation of the alloying elements because we want them to be dissolved in the intermetallic matrix.

## 3. Results and Discussion

Firstly, we create in the computational cell the completely disordered state by assigning the sorts of the atoms Ni, Al or Cr randomly with the probability equal to their concentrations. We do this for EP975 and VKNA-25 alloys taking the concentrations of the elements from Table 2 (in at.%). Then one vacancy is created in the computational cell and the vacancy diffusion algorithm described in Section 2.3 is used to model the equilibration of the alloy structure. The simulation is conducted at the temperature $T = 1200\,^{\circ}\text{C}$ until the total energy of the alloy reaches the minimal value. The minimal energy value was achieved in $10^{10}$ jumps of the vacancy. This way, the equilibrium structures of EP975 and VKNA-25 alloys were obtained.

Secondly, in the half of the computational cell, $1 \le n_y \le N_y/2$ equilibrium EP975 alloy is placed, while in the second half, $(N_y/2) + 1 \le n_y \le N_y$, we place the equilibrium VKNA-25 alloy. Again, single vacancy is placed in the computational cell and the interdiffusion of the components between the two alloys is simulated at the temperature $T = 1200\,^{\circ}\text{C}$. The duration of this simulation is $3 \times 10^{11}$ vacancy jumps.

In Figure 3, we present the evolution of the structure of the alloys during diffusion bonding. In (a), the initial structure at $t = 0$ is shown. In the lower (upper) half of the computational cell, EP975 (VKNA-25) alloy is found. The inset shows the atomic resolution of the structure. Dark blue and light blue colors are used for Ni and Al atoms, respectively. They form ordered structure with Ni$_3$Al stoichiometry. Alloying elements (Cr) appear in yellow color. They are dissolved in the intermetallic matrix forming small clusters. In Figure 3b,c structure is shown at $t = 4.5 \times 10^{10}$ and $3.0 \times 10^{11}$ (note that the number of vacancy jumps is the measure of time in our simulations). Gradual redistribution of the alloy elements can be observed.

Time evolution of the concentration profiles is presented in Figure 4a–c for Ni, Al and Cr, respectively, in at.%. Black curves 1 stand for the initial distribution of components, while red curves 2 and blue curves 3 are for $t = 4.5 \times 10^{10}$ and $3.0 \times 10^{11}$, respectively. Since periodic boundary conditions are used, we have periodic structure with alternating layers of EP975 and VKNA-25 alloys. It can be seen that Ni and Al move from VKNA-25 into EP975, while Cr in the opposite direction. As a result, the distribution of elements becomes homogeneous. This behavior is in qualitative agreement with the experimental results presented in Figure 2.

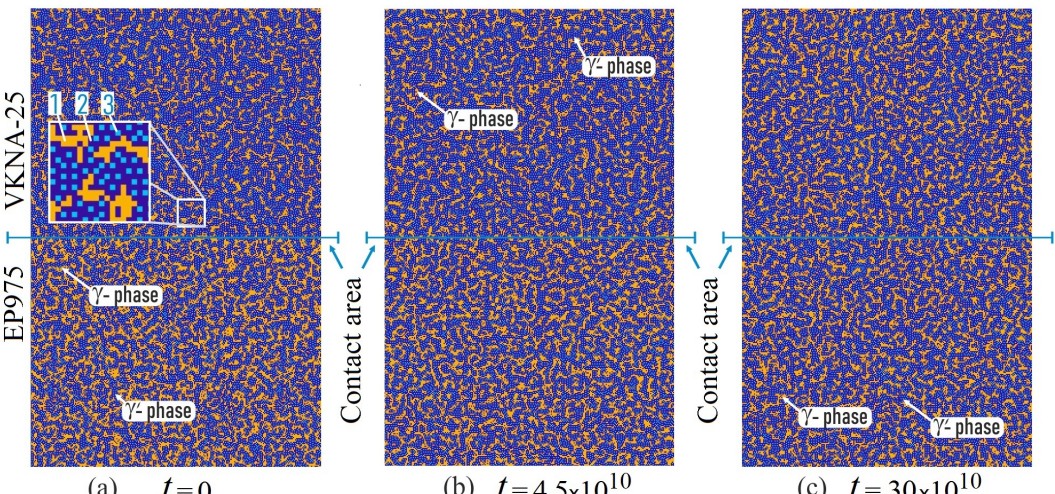

**Figure 3.** Structure evolution during diffusion bonding of EP975 and VKNA-25 alloys. (**a**) Initial structure with the lower (upper) half occupied by the EP975 (VKNA-25) alloy. Inset shows the structure at the atomic scale. Ni, Al and Cr are shown by dark blue, light blue and yellow colors. Ni and Al create Ni$_3$Al intermetallide and Cr is dissolved in it creating small clusters. (**b**,**c**) Structure at $t = 4.5 \times 10^{10}$ and $3.0 \times 10^{11}$, respectively. The number of vacancy jumps is used as the measure of time.

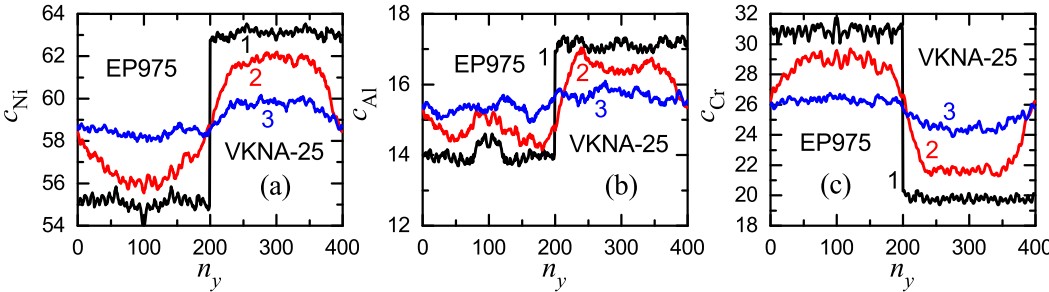

**Figure 4.** Distribution of concentrations of elements in at.% along the $y$ direction for (**a**) Ni, (**b**) Al and (**c**) Cr. The concentration profiles are presented for $t = 0$ (black lines 1), $t = 4.5 \times 10^{10}$ (red lines 2) and $t = 3.0 \times 10^{11}$ (blue lines 3).

Order parameters at the first coordination shell, calculated with the help of Equation (10), are shown as the functions of time in Figure 5. Only $\alpha^{(1)}_{\mathrm{NiCr}}$ shows a noticeable change in time, while other two order parameters remain practically constant. This is explained by our choice of the interatomic potential parameters, see Equation (19). Parameter $D_{\mathrm{NiCr}}$ is smaller than $D_{\mathrm{NiAl}}$ and $D_{\mathrm{AlCr}}$. That is why the reconstruction of Ni-Cr bonds takes place with a higher probability during the diffusion process.

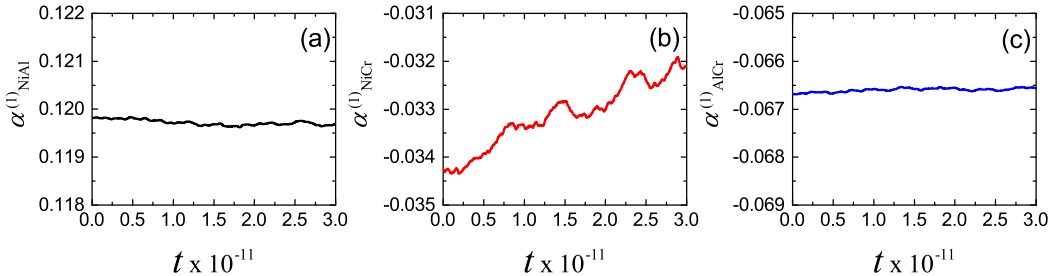

**Figure 5.** Time evolution of order parameters at the first coordination shell: (**a**) $\alpha^{(1)}_{\mathrm{NiAl}}$, (**b**) $\alpha^{(1)}_{\mathrm{NiCr}}$ and (**c**) $\alpha^{(1)}_{\mathrm{AlCr}}$.

The absolute value of the order parameter $\alpha^{(1)}_{\mathrm{NiAl}}$ is considerably larger than that of $\alpha^{(1)}_{\mathrm{NiCr}}$ and $\alpha^{(1)}_{\mathrm{AlCr}}$, since Ni and Al form an ordered phase $\mathrm{Ni_3Al}$.

There is a relation between simulation time and temperature as regards the kinetics of welding and order parameter evolution. With an increase in temperature, the probabilities in Equation (16) become closer, the ordering process becomes faster but the maximal values of order parameters become smaller. In addition, with an increase in temperature, the rate of vacancy jumps increases.

## 4. Conclusions

A theoretical basis and its numerical implementation were developed for modeling diffusion processes in refractory nickel-base alloys. As an example, diffusion bonding of VKNA-25 and EP975 alloys is considered. All alloy components were classified into three groups as explained in Section 2.4, which made it possible to consider multi-component alloys as ternary.

In Section 2.2, expression for the alloy energy, Equation (6), was given under the assumption of non-deformable lattice and pairwise interatomic interactions. The difference between alloy energies in any state and in the disordered state, $\Delta E_1$, is expressed in terms of the order parameters $\alpha^{(i)}_{KL}$ and ordering energies $\omega^{(i)}_{KL}$ [see Equation (9)]. On the other hand, the difference between alloy energies in any state and in the decomposed into pure components state, $\Delta E_2$, is expressed in terms of the parameters $\beta^{(i)}_{KL}$ and ordering energies $\omega^{(i)}_{KL}$ (see Equation (12)). This means that the parameters $\beta^{(i)}_{KL}$ determine the closeness of the alloy to decomposition into pure components.

In Section 2.3, the vacancy diffusion model was described. The model takes into account the effect of temperature and the energy gain/loss due to the vacancy jump. A vacancy jump leading to an increase in the potential energy of the alloy is possible, but its probability is less than the probability of a jump leading to a decrease in the potential energy. As the temperature rises, the probability that atoms of different sorts will jump into a vacancy is getting closer. This leads to a transition to a disordered state at a sufficiently high temperature.

It is important to note that all expressions presented in Sections 2.2 and 2.3 are valid for ternary alloys based on any lattice of any dimension. Information on the type and dimension of the lattice is taken into account by the coordination numbers $N_i$.

Overall, a model of ternary alloys was developed that allows to simulate diffusion processes in heat-resistant Ni-base alloys.

In this work, a toy model of refractory alloys based on a two-dimensional square lattice was used. In an upcoming study, the developed model will be applied to the simulation of diffusion bonding of refractory alloys using a more realistic three-dimensional fcc lattice.

**Author Contributions:** Conceptualization, V.A.V. and S.V.D.; methodology, V.A.V., E.A.K. and S.V.D.; software, A.R.K. and E.A.S.; experimental data E.V.G. and V.A.V.; investigation, A.R.K. and E.A.S.; data curation, A.R.K. and E.A.S.; writing—original draft preparation, S.V.D. and E.A.K.; visualization, A.R.K. and E.A.S. All authors have read and agreed to the published version of the manuscript.

**Funding:** This research was funded by the Russian Science Foundation, grant No. 18-19-00685.

**Acknowledgments:** This work was partly supported by the State Assignment of IMSP RAS No. AAAA-A17-117041310220-8.

**Conflicts of Interest:** The authors declare no conflict of interest.

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
