# Peer review of "Simulation of Diffusion Bonding of Different Heat Resistant Nickel-Base Alloys"

_computation, doi:10.3390/computation8040102_

Round 1
Reviewer 1 Report
- Line 19-21: It is mandatory to explain the reason for the citation of each one of the referred papers (or small grouped references two or max three). Please, include one line explaining the technical reason for the quotation. Moreover, avoid grouped references as done, i.e. : 1-12 or 13-21. Please consider this as a major concern.
- In Tab.I: please include Ni, to be indicated as Bal (Balance)
- In the caption of Fig.1 the explanation of (c ) microstructure is missing
- In your Fig 2: are presented the diffusion elements profiles across the contact surface. The diffusion bonding phenomenon is active from -20 to + 20 microns, according to your observation. Please indicate if this profile is an EDS profile, indicate the instrument used.
- After the profile reported in Fig.2 I was expecting another graph with calculated elements concentration versus distance from the interface. Please, add this graph with a discussion about precision and any correlation that can be outlined.
Author Response
We would like to thank the reviewer for reading our manuscript and for suggestions for improving it.
Reviewer wrote:
- Line 19-21: It is mandatory to explain the reason for the citation of each one of the referred papers (or small grouped references two or max three). Please, include one line explaining the technical reason for the quotation. Moreover, avoid grouped references as done, i.e. : 1-12 or 13-21. Please consider this as a major concern.
Our response:
We have extended the Introduction and gave a description for small groups of references. In particular, the sentence
“The theory and practice of diffusion welding of dissimilar metals has been actively developed over the past decades due to the undoubted importance of this process for various technologies [1-12].”
was replaced with
“The theory and practice of diffusion welding of dissimilar metals has been actively developed over the past decades due to the undoubted importance of this process for various technologies [1,2]. Some alloys can be successfully bonded without the use of an interlayer [3-5]. However, in most cases, to avoid the formation of brittle intermetallic inclusions, an interlayer metal is used. Examples are bonding of titanium to steel [6-9], nickel [10], copper [11], and the use of copper interlayer for bonding stainless steel to Zircaloy 4 [12].”
The sentence
“Heat-resistant intermetallic alloys can have unique physical and mechanical properties (heat resistance, resistance to oxidation, corrosion, etc.) [13-21].”
was replaced with
“Heat-resistant intermetallic alloys can have unique physical and mechanical properties (heat resistance, resistance to oxidation, corrosion, creep performance, etc.) that makes them ideal for a number of applications in aerospace industry [13-17]. Structure and properties of Ni-base alloys can be improved by laser treatment [18,19] and thermomechanical treatment [20,21]. Electro-discharge machining process of Inconel-718 superalloy has been optimized in [22].”
The sentence
“Methods for joining metals and superalloys include diffusion welding, diffusion welding through a molten layer, fusion welding, friction welding, diffusion welding in a liquid phase, friction stir welding, etc. [17,22-29].”
was replaced with
“Methods for joining metals and superalloys include diffusion welding [15,23], fusion welding, friction welding [24-28], diffusion welding in a liquid phase [29], friction stir welding [28], etc.”
Reviewer wrote:
- In Tab.I: please include Ni, to be indicated as Bal (Balance)
Our response:
This addition was made.
Reviewer wrote:
- In the caption of Fig.1 the explanation of (c ) microstructure is missing
Our response:
In the caption of Fig. 1 we have added:
“(c) Microstructure of the joint ofEP975 (bottom) and VKNA-25 (top) alloys.”
Reviewer wrote:
- In your Fig 2: are presented the diffusion elements profiles across the contact surface. The diffusion bonding phenomenon is active from -20 to + 20 microns, according to your observation. Please indicate if this profile is an EDS profile, indicate the instrument used.
Our response:
The following text was added in relation to Fig. 2:
“The distribution of elements in the solid-phase joint zone was determined by the X-ray energy-dispersive spectroscopy (EDS) on a Tescan VEGA 3SBH scanning electron microscope with an EDX attachment. The measurements were carried out only from the γ-phase, moving from the EP975 alloy to the VKNA-25 alloy with a step of 10 μm. The diffusion zone was found to be about 35 μm wide. To plot the distribution of alloying elements, the EDS data were averaged over at least three measurements per point.”
In relation to Fig. 1 we have added:
“The microstructures of the alloys and the joint zone were studied by scanning electron microscope Mira 3LMH (TESCAN, Czech Republic).”
Reviewer wrote:
- After the profile reported in Fig.2 I was expecting another graph with calculated elements concentration versus distance from the interface. Please, add this graph with a discussion about precision and any correlation that can be outlined.
Our response:
This is indeed so. We have added paned (b) to Fig. 2 and the following text related to the new plot:
“In Fig. 2(b), the concentrations of three groups of alloy components are shown: the base element Ni, the intermetallic-forming Al and Ti are combined and named Al, and all other elements are combined and named Cr. The merit of combining alloy elements into these three groups is explained in Sec. 2.4.”
Overall, we hope that the reviewer finds the above corrections sufficient.
Reviewer 2 Report
In this work Albert R. Khalikov and co-workers reported the simulation of diffusion bonding of VKNA-25 and EP975 alloys by developing a theoretical basis and its numerical implementation for modeling diffusion processes. Staring from the idea that alloy components can be divided into the base element Ni, the intermetallic forming elements Al and Ti, and the alloying elements, the authors performed calculations within the framework of the hard sphere model in terms of order parameters and ordering energies, and used a vacancy diffusion model to describe diffusion process. Then presented and analyzed the evolution of the structure of the alloys during diffusion bonding.
In general, this is a very important scientific contribution, particularly from the development of computational methods that allows to simulate diffusion processes in heat-resistant Ni-base alloys, admittedly, the model based on three-dimensional fcc lattice would be more realistic. Additionally, the paper is well structured, exhibiting a valuable set of conclusions properly supported by the work performed throughout the text. I therefore recommend this work to be considered for acceptance in Computation.
Author Response
We thank the reviewer for evaluating our manuscript and were pleased with the positive feedback.
Reviewer 3 Report
Author:
1) During the welding or joining process if the interface is perpendicular to the direction of pressure or if its parallel to direction of pressure will there be different in inter phase diffusion or void creation.
2) In pressure welding is there a relationship between time, temperature and pressure applied on the interphase development (with reference to figure 5)?
3) Is rate of change in temperature factor for vacancy jump? Is there a relationship with pressure?
4) Bonding temperature can significantly affect diffusion in interfacial zone. Microstructures formed diffusion and vacancy jump, are they dependent on the temperature and rate of solidification?
(5) Are there any effects of liquid/semi solid phase formed on diffusion and vacancy jump at interfacial region during bonding process?
Author Response
We wish to thank the Reviewer for reading the manuscript and for rising some questions and critical comments. Below we give our point by point response to all of them and describe amendments applied to the manuscript.
Reviewer wrote:
1) During the welding or joining process if the interface is perpendicular to the direction of pressure or if its parallel to direction of pressure will there be different in inter phase diffusion or void creation.
Our response:
Indeed, this point was missing in our description of experimental setup. The following sentence was added to the experimental details:
“A compressive strain is applied normal to the interface to achieve better physical contact between the parts to be welded.”
Reviewer wrote:
2) In pressure welding is there a relationship between time, temperature and pressure applied on the interphase development (with reference to figure 5)?
Our response:
The following missing information was added to the description of experimental procedure:
“Note that VKNA-25 is a hard-to-deform single-crystal alloy, while EP975 is polycrystalline alloy that flows superplastically at the selected temperature and strain rate. The superplasticity of EP975 alloy is important because it helps to achieve a better physical contact during pressure welding and to minimize the formation of voids. Strain control loading is performed so that pressure is not a controlled parameter. A compressive strain is applied normal to the interface to achieve better physical contact between the parts to be welded.”
In relation to Fig. 5, at the end of Sec. 3, the following text was added:
“There is a relation between simulation time and temperature as regards the kinetics of welding and order parameter evolution. With an increase in temperature the probabilities in Eq. (16) become closer, the ordering process becomes faster but the maximal values of order parameters become smaller. Also with increase in temperature the rate of vacancy jumps increases.”
Reviewer wrote:
3) Is rate of change in temperature factor for vacancy jump? Is there a relationship with pressure?
Our response:
Yes, as mentioned in the response to item 2), with increase in temperature the rate of vacancy jumps increases. The vacancy diffusion model described in Sec. 2.3 takes into account temperature, but pressure is not taken into account, because we use the rigid lattice assumption without taking into account the effect of atomic relaxation. The following clarification is added in the end of Sec. 2.3:
“Pressure is not taken into account in this model, because we use the rigid lattice assumption without taking into account the effect of atomic relaxation.”
Reviewer wrote:
4) Bonding temperature can significantly affect diffusion in interfacial zone. Microstructures formed diffusion and vacancy jump, are they dependent on the temperature and rate of solidification?
Our response:
It is true that the bonding temperature significantly affects diffusion in interfacial zone. Actually, in our experiments and in our model, we do not have solidification because joining takes place in the solid state without melting. The high temperature activates the diffusion and superplastic flow of the EP975 alloy, and these two factors play a positive role in achieving a high-quality joint. The following text was added in Sec. 2.4, third paragraph:
“In our experiments and in our model, we do not have solidification because joining takes place in the solid state without melting. The high temperature activates the diffusion and superplastic flow of the EP975 alloy, and these two factors play a positive role in achieving a high-quality joint.”
Reviewer wrote:
(5) Are there any effects of liquid/semi solid phase formed on diffusion and vacancy jump at interfacial region during bonding process?
Our response:
As described in the reply to item 5), joining of VKNA-25 and EP975 superalloys takes place in the solid state without melting.
Once again, we thank the Reviewer for the criticism that helped us to improve presentation of our model and simulation results and hope that the Reviewer finds our revised manuscript suitable for publication in Computation.
Reviewer 4 Report
In this manuscript, the author presents a theoretical model that allows one two study the diffusion process in nickel base refractory alloys through divide the alloy components into three groups. This approach allows one to consider multicomponent alloys as ternary alloys, which greatly simplifies the analysis. Diffusion bonding of EP975 and VKNA-25 are modeled, respectively. The author claims that the results obtained indicated that the ternary alloy model is efficient in modeling the diffusion of dissimilar Ni-base refractory alloys. However, I think the research in this article lacks completeness because these results are not compared with existing methods.
Other comments:
- As claimed by author, the main difficulty in atomistic simulation of heat-resistant Ni-base alloys is that there are too many kinds of alloy elements to construct interatomic potentials. However, the authors divide the alloy components into three groups roughly and ignore the detailed chemical composition of two alloys. It is hard to say that this approach is reasonable;
- This proposed method should be compared with existing methods such as MD or continuum model to verify its correctness;
- The size of the computational cell is , and the PBC is employed, but as depicted by Fig 2, the width of the interlayer affected by the diffusion is about 20 , I think there should be more explanation for the scale issue.
- Some typos:
- line 84, A repeated sentence.
- The line above Eq. (1), what do you mean the concentrations of atoms of types A, B, and C are equal XXX?
- Line 188, Secs.
Author Response
Thanks to the Reviewer for reading our manuscript and for critical comments. We have done our best to address them as described below.
Reviewer wrote:
- As claimed by author, the main difficulty in atomistic simulation of heat-resistant Ni-base alloys is that there are too many kinds of alloy elements to construct interatomic potentials. However, the authors divide the alloy components into three groups roughly and ignore the detailed chemical composition of two alloys. It is hard to say that this approach is reasonable;
Our response:
We understand that our model is crude, but we hope that it can be useful because it provides an opportunity to approach a problem that would otherwise appear intractable. We have added the following text to the first paragraph of Sec. 2.4. Ternary alloy model and simulation setup:
“It should be noted that the ternary alloy model can describe the considered multicomponent superalloys only at a qualitative level.”
Reviewer wrote:
- This proposed method should be compared with existing methods such as MD or continuum model to verify its correctness;
Our response:
In our work we provide a qualitative comparison of the numerical results with the experimental results. In principle, it is possible to use the MD method, but it is difficult to apply it to simulate the diffusion process, since the time scale achieved by this method is on the order of 1 ns, and diffusion bonding takes about 15 min. The timescale difference is huge and amounts to twelve orders of magnitude. This is why the development of the Monte Carlo method is very important. The use of continuum models is not straightforward, since it requires knowledge of the matrix of diffusion coefficients, which can be obtained using microscopic methods (MD or Monte Carlo). We plan to do this in future studies.
The following text was added in the Introduction, where we characterize different simulation methods:
“It is difficult to apply MD to simulate the diffusion process, since the time scale achieved by this method is on the order of 1 ns, and diffusion bonding takes about 15 min. The time difference is huge, twelve orders of magnitude. This is why the development of the Monte Carlo method is very important. The use of continuum models for diffusion in multi-component media is not straightforward, since it requires knowledge of the matrix of diffusion coefficients [39].”
Reviewer wrote:
- The size of the computational cell is , and the PBC is employed, but as depicted by Fig 2, the width of the interlayer affected by the diffusion is about 20 , I think there should be more explanation for the scale issue.
Our response:
The following text was added in Sec. 2.4 to the discussion of the 2D model used in our simulations:
“The typical interatomic distance in metals is about 2.5 A, then the size of the computational cell is about 0.1 mm. This is two orders of magnitude smaller than the experimentally observed diffusion zone of 35 mm, see Fig. 2. An increase in the size of the computational cell requires an increase in the simulation time.”
Reviewer wrote:
- Some typos:
- line 84, A repeated sentence.
- The line above Eq. (1), what do you mean the concentrations of atoms of types A, B, and C are equalXXX?
- Line 188, Secs.
Our response:
- We have removed the repeated sentence “Let us consider a three-component ABC alloy, in which atoms of types A, B, and C are located at lattice sites.”
- The sentence was corrected as follows: “The concentrations of atoms of types A, B, and C are equal to, respectively,”
- We have replaced “in Secs. 2.3 and 2.2” with “in Sec. 2.2 and Sec. 2.3”.
We hope the Reviewer finds the above corrections and arguments satisfactory.
Round 2
Reviewer 1 Report
Thank you for having taking into account the suggestions
Reviewer 4 Report
the authors have adressed all my previous concerns, and along with other reviewer's comment, i keep no resevation on its publication on this journal